# Orthogonal Functionalization of Nanodiamond Particles after Laser Modification and Treatment with Aromatic Amine Derivatives

**DOI:** 10.3390/nano8110908

**Published:** 2018-11-05

**Authors:** Justyna Fraczyk, Adam Rosowski, Beata Kolesinska, Anna Koperkiewcz, Anna Sobczyk-Guzenda, Zbigniew J. Kaminski, Mariusz Dudek

**Affiliations:** 1Institute of Organic Chemistry, Lodz University of Technology, Zeromskiego 116, 90-924 Lodz, Poland; justyna.fraczyk@p.lodz.pl (J.F.); beata.kolesinska@p.lodz.pl (B.K.); zbigniew.kaminski@p.lodz.pl (Z.J.K.); 2SPI Lasers, 3 Wellington Park, Tollbar Way, Hedge End, Southampton, Hampshire SO30 2QU, UK; adam.rosowski@spilasers.com; 3Institute for Manufacturing, University of Cambridge, 17 Charles Babbage Road, Cambridge CB3 0FS, UK; 4Institute of Materials Science and Engineering, Lodz University of Technology, Stefanowskiego 1/15, 90-924 Lodz, Poland; a.koperkiewicz1991@gmail.com (A.K.); asobczyk@p.lodz.pl (A.S.-G.)

**Keywords:** nanodiamond powders, laser treatment, aromatic amine, diazonium salt, isokinetic mixture, amphiphilic NDPs

## Abstract

A laser system with a wavelength of 1064 nm was used to generate sp^2^ carbon on the surfaces of nanodiamond particles (NDPs). The modified by microplasma NDPs were analysed using FT-IR and Raman spectroscopy. Raman spectra confirmed that graphitization had occurred on the surfaces of the NDPs. The extent of graphitization depended on the average power used in the laser treatment process. FT-IR analysis revealed that the presence of C=C bonds in all spectra of the laser-modified powder. The characteristic peaks for olefinic bonds were much more intense than in the case of untreated powder and grew in intensity as the average laser power increased. The olefinized nanodiamond powder was further functionalized using aromatic amines via in situ generated diazonium salts. It was also found that isokinetic mixtures of structurally diverse aromatic amines containing different functional groups (acid, amine) could be used to functionalize the surfaces of the laser-modified nanoparticles leading to an amphiphilic carbon nanomaterial. This enables one-step orthogonal functionalization and opens the possibility of selectively incorporating molecules with diverse biological activities on the surfaces of NDPs. Modified NDPs with amphiphilic properties resulting from the presence carboxyl and amine groups were used to incorporate simultaneously folic acid (FA-CONH-(CH_2_)_5_-COOH) and 5(6)-carboxyfluorescein (FL-CONH-(CH_2_)_2_-NH_2_) derivatives on the surface of material under biocompatible procedures.

## 1. Introduction

Diamonds have for centuries been associated with luxury jewellery, but since the development of artificial manufacturing techniques, they have become relatively inexpensive and readily available [1]. This has created the opportunity for new and wider applications of this extremely versatile material. In particular, the development of nano-scale diamond powders has opened new research avenues in medicine, stimulated by their good biocompatibility in tissue environments, biofunctionality, and non-toxicity [2,3]. The unique properties of nanodiamond particles (NDPs) have also been exploited for the transport of pharmaceutical and diagnostic substances [2,4,5,6,7,8].

Currently, extensive research is focused on functionalization of diamond powders [9,10]. Nanodiamonds [11] are nanoparticles with diameters in the range of 2–20 nm. They consist of crystal domains with diamond-like carbon atoms arrangement in a cubic lattice (due to the sp^3^ hybridization of the carbon atom). The methods of functionalizing nanodiamond are different from those used for modifying fullerenes, carbon nanotubes, or graphene formed by sp^2^ hybridized carbon atoms. The most common method for functionalizing nanodiamond is the redox reaction, which results in a change in the oxidation state of the surface atoms. Oxidation can also be used to increase the number of oxygen atoms on nanodiamond surfaces. The most commonly used method of introducing hydroxyl groups onto the surfaces of NDPs is Fenton’s reaction [12], which uses Fe^2+^ salts and H_2_O_2_. The introduction of carboxyl groups can be accomplished using various other oxidants. The most popular are mixtures [13] of H_2_SO_4_ and HNO_3_ (3:1, v:v) or of sulphuric, nitric, and perchloric acid. Use of piranha water leads to the introduction of carboxyl groups on NDP surfaces. Due to the substantially higher reactivity of sp^2^ carbon compared to sp^3^ carbon, the oxidation of nanodiamonds with oxidizing acids leads to purification of the nanodiamond powder and to the removal of sp^2^ components. Thus, oxidation also increases the phase purity [14].

Nanodiamonds containing hydroxyl and carboxy groups are the most commonly used functionalized nanomaterials in medicine. The presence of carboxyl groups enables the formation of stable amide (peptide) bonds and the covalent ester linking of functionalizing compounds to the nanodiamond surface. Amino-functionalized nanodiamonds are synthesized from hydroxyl-functionalized NDPs with (3-aminopropyl)trimethoxysilane [15]. Another group of chemically functionalized nanodiamonds comprises materials with fluorine or chlorine atoms on their surfaces. Fluorine is introduced in a reaction with an F_2_/H_2_ mixture at an elevated temperature [16]. The properties of nanodiamonds can be further modulated (to become similar to those of CNTs, fullerene, or graphene) by introducing sp^2^ carbon atoms onto their surfaces [17]. Small amounts of sp^2^ carbon can be formed on NDP surfaces [18] through careful thermal annealing at 750 °C *in vacuum* and via the mechanochemical formation of sp^2^ carbon-related dangling bonds on the NDPs by intense ultrasonication.

In recent years, laser technologies have become of great importance for the surface treatment of bulk diamond materials. Laser technologies enable the microplasma modification of diamonds at very specific points, changing their physical and mechanical properties in certain places, in different ways (see for example [19,20,21]). Laser ablation also is used to manufacture carbon-based nanomaterials [22,23]. Thus, laser technologies are contributing significantly to the development of carbon-based nanomaterials, including NDPs, which have a growing range of applications in many fields of science and technology.

The aim of this study was to obtain a nanodiamond powder containing double carbon–carbon bonds on the surface using a laser system allowing microplasma modification of powder surface instead of the standard thermal process. The effectiveness of the process of laser treatment and chemical modification was monitored using Raman and FT-IR spectroscopy. The usefulness of the obtained olefinized (with sp^2^ carbon atoms) nanodiamond powder was evaluated in a reaction with aromatic amines in the presence of isoamyl nitrite, in a process based on the in situ generation of diazonium salts. We also investigated the possibility of using isokinetic mixtures of two different aromatic amines as substrates for the functionalization of the carbon nanomaterials. This approach would allow the simultaneous addition of aromatic derivatives with different functional groups in the same, similar, or preferred ratio of the concentrations on the surfaces of the NDPs, for full orthogonal functionalization of the nanomaterial. It would also significantly reduce the number of steps and the time needed for modification of the nanomaterial with a variety of substituents (such as markers, drugs, or homing ligands).

## 2. Materials and Methods

Nanodiamond powders with purity above 98% (low contents of metallic contamination) were purchased from SIGMA-ALDRICH (Poznan, Poland). The spherical particles had a diameter of about 4 nm and a specific surface area ranging from 300 to 400 m^2^/g.

The NDPs were modified using a 20W G3 SPI pulsed laser system (SPI Lasers UK Ltd., Southampton, UK) with a wavelength of 1064 nm and a line band width of <4 nm. The laser system pulse duration was in the range of 10–200 ns with an operating frequency in the region of 1–500 kHz, depending on the pulse duration used. The laser beam was delivered to a GSI lightning scanner head with a Linos F-Theta 100 mm focusing lens (Figure 1). The beam quality, M^2^ of the laser in single mode, was <1.6. The diameter of the laser spot on the substrate was about 30 μm. The scanner head with an F-Theta lens enabled the beam to be moved over the substrate (surface of spilled powder) while still in focus. The powder (200 mg) was placed in a glass ampoule closed with a polymer stopper. A scanning area of 14 × 6 mm was processed 25 times in one sample arrangement. The sample was then shaken to mix the powder in the ampoule before the scanning process was repeated. The procedure was repeated once more, resulting in a total of 3 × 25 = 75 repetitions of the scanning process. Each cycle of laser treatment was carried out with the parameters shown in Table 1.

Raman spectroscopy analysis: Samples of the nanodiamond powders before and after laser modification were placed on microscope slides for analysis by Raman spectroscopy. A Renishaw inVia Raman Microscope (Gloucestershire, UK) was used, equipped with a 532 nm laser arranged in a backscattering geometry. The power of the laser was 0.4 mW. The investigated wavenumber ranged from 100 cm^−1^ to 3200 cm^−1^. All measurements were carried out at room temperature and in air atmosphere. For more accurate positioning of the bands, the Raman spectra were deconvoluted using the Lorentzian line function.

FT-IR spectroscopy analysis: The chemical and physical structures of the materials were investigated by Diffuse Reflectance Infrared Fourier Transform Spectroscopy (DRIFTS) using the Thermo Scientific Nicolet iS50 FT-IR spectroscope (Waltham, MA, USA). DRIFT spectra were collected in the range of 400–4000 cm^−1^. Before measurements, the samples were mixed with potassium bromide (KBr) (Sigma-Aldrich, Poznan, Poland).

SEM microscopy analysis: Microscopic examinations of topography of powder surfaces were carried out using TECAN VEGA3 scanning electron microscope equipped with an electronic optics system that allows observation of the surface of various materials at magnifications from 4–1,000,000 times, in the energy range of the incident electron beam from 0.2 to 30 keV with a maximum resolution of 3 nm in the secondary emission electron mode and 3.5 nm in the backscattered electron mode of operation. SEM microscopic images of surface topography were made using high vacuum mode with 20 keV probing energy. The surface of each sample was sprayed with a gold acting as a conductive substance using a Quorum Technologies Ltd (Lewes, UK). vacuum sprayer. The magnifications 300×, 1000×, 5000×, 10,000×, and 20,000× were used to study the surface topography.

### 2.1. Chemical Functionalization of Nanodiamond Powder with Olefin Bonds on the Surface

#### 2.1.1. Attachment of Benzoic Acid Derivative to the Surface of the Nanodiamond Powder (**1**)

The laser-modified nanodiamond powder with C=C bonds (50 mg), prepared as described above, was suspended in 1 N HCl (10 mL) and sonicated for 15 min in an ultrasonic bath. The suspension was then heated to 80 °C with vigorous stirring. In the next step, p-aminobenzoic acid (0.686 g, 5 mmol) and isoamyl nitrite (0.7 mL, 5 mmol) were added. The sample was heated at 80 °C with vigorous stirring for another 18 h. The suspension was diluted with distilled water (50 mL) and filtered. The precipitate was washed with a mixture of DMF–water (1:1) (20 mL), DMF (20 mL), and DCM (20 mL). Washing was continued until no reagents were found in the filtrate (TLC control). The residue was suspended in water (10 mL) and lyophilized.

#### 2.1.2. Synthesis of Aminonaphtalene Derivative Linked to the Surface of the Nanodiamond Powder (**2**)

The nanodiamond powder with C=C bonds (50 mg) was suspended in 1 N HCl (10 mL) and sonicated for 15 min. The suspension was heated to 80 °C with vigorous stirring, and 1,5-diaminonaphthalene (0.791 g, 5 mmol) and isoamyl nitrite (0.7 mL, 5 mmol) were then added. Stirring and heating at 80 °C was continued for 18 h. The suspension was diluted with distilled water (50 mL) and filtered. The precipitate was washed with a mixture of DMF–water (1:1) (20 mL), DMF (20 mL), and DCM (20 mL). Washing was continued until no reagents were found in the filtrate (TLC control). The residue was suspended in water (10 mL) and lyophilized.

#### 2.1.3. Coupling of 6-Aminohexanoic Acid Methyl Ester to Benzoic Acid Derivative on Nanodiamond Powder Surface (**3**)

Nanodiamond powder modified with benzoic acid **1** (30 mg) was suspended in a mixture of DCM and DMF (1:1) (10 mL). The suspension was cooled in a water-ice bath to 0 °C. DMT/NMM/TosO^−^ and NMM (110 µL, 1 mmol) were then added with vigorously stirring. Stirring was continued for 60 min. Next, methyl ester of 6-aminohexanoic acid hydrochloride (0.363 g, 2 mmol) and a stoichiometric amount of NMM (220 μL, 2 mmol) were added to the suspension. Stirring was continued for 24 h at room temperature. The precipitate was filtered off under reduced pressure and washed with DCM (10 mL), DMF (10 mL), DMF–water (1:1) (10 mL), DMF (10 mL), and again with DCM (10 mL). Washing was continued until no reagents were found in the filtrate (TLC control). The residue was suspended in water (10 mL) and lyophilized.

#### 2.1.4. Synthesis of Naphthylamine Derivative with Ketoprofen Attached to the Nanodiamond Powder Surface (**4**)

Ketoprofen (0.508 g, 2 mmol) was dissolved in a mixture of DCM and DMF (1:1) (10 mL). After cooling the solution to 0 °C, DMT/NMM/TosO^−^ (0.826 g, 2 mmol) and NMM (110 μL, 1 mmol) were added with vigorous stirring. Stirring was continued for 1 h until the reagent was consumed. Once activation was complete, nanodiamond powder **2** functionalized with 1,5-diaminenaphtalene (30 mg) and NMM (220 μL, 2 mmol) was added to the solution of Ketoprofen triazine ester. The suspension was stirred for 12 h at room temperature. The precipitate was filtered off under reduced pressure and washed with DCM (10 mL), DMF (10 mL), water: DMF (1:1) (10 mL), DMF (10 mL), and again with DCM (10 mL). Washing was continued until no reagents were found in the filtrate (TLC control). The residue was suspended in water (10 mL) and lyophilized.

#### 2.1.5. Synthesis of Nanodiamond Derivative (**5**) Containing Both Benzoic Acid Residue and Naphthylamine on the Surface

The nanodiamond powder with C=C bonds (50 mg) was suspended in 1 N HCl (10 mL) and sonicated for 15 min. The suspension was then heated to 80 °C with vigorous stirring. In the next step, p-aminobenzoic acid (0.686 g, 5 mmol), 1,5-diaminonaphthalene (0.791 g, 5 mmol), and isoamyl nitrite (1.4 mL, 10 mmol) were added. Intensive stirring and heating at 80 °C were continued for 18 h. The suspension was diluted with distilled water (50 mL) and filtered. The precipitate was washed with a mixture of DMF–water (1:1) (20 mL), DMF (20 mL), and DCM (20 mL). Washing was continued until no reagents were found in the filtrate (TLC control). The residue was suspended in water (10 mL) and lyophilized.

#### 2.1.6. Synthesis of Nanodiamond Derivative (**6**) Containing Folic Acid Residue and 5(6)-Carboxyfluorescein Orthogonally Attached to the Surface via Two Different, Amphiphilic Linkers

The nanodiamond derivative (**5**) containing both benzoic acid residue and naphthylamine on the surface (50 mg) were suspended in DMF (5 mL). The suspension was sonicated for 10 min before the addition of 4-(4,6-dimethoxy-[1,3,5]triazin-2-yl)-4-methylmorpholinium tetrafloroborate (DMT/NMM/BF_4_^−^; 0.325 g, 1 mmol) and *N*-methylmorpholine (NMM; 0.11 mL, 1 mmol) in DMF (2 mL). The mixture was sonicated for 1 h at 10 °C. The synthetized triazine ester, immobilized on the surface of the modified NDP, was treated with 5(6)-carboxyfluorescein derivative FL-CONH-(CH_2_)_2_-NH_2_ (0.418 g, 1 mmol). The suspension was stirred vigorously using a magnetic stirrer for 12 h at room temperature. The functionalized NDP was then filtered and washed three times with DMF (5 mL), a mixture of water and DMF (1:1; 5 mL), water (5 mL), and finally DMF (5 mL) to remove by-products of the reaction and excess reagents. The functionalized NDP was lyophilized. Lyophilized NDPs modified by attachment of FL-CONH-(CH_2_)_2_-NH_2_ (50 mg) were suspended in 2 mL of DMF. The suspension was sonicated for 30 min. Meanwhile, a mixture of folic acid derivative FA-CONH-(CH_2_)_5_-COOH (0.554 g; 1 mmol), DMT/NMM/TosO^−^ (0.413 g; 1 mmol) and NMM (0.22 mL; 2 mmol) in DMF (5 mL) was prepared, cooled to 5 °C, and stirred for 45 min in the dark. After 45 min, the mixture was added to the suspended NDPs. The reaction mixture was sonicated for additional 45 min at 10 °C in the dark. Stirring was continued for 24 h. The product **6** was filtered and washed with 25% NMM in DMF (5 mL), with DMF (3 × 5 mL), and with a mixture of DMF and water (1:1; 3 × 5 mL). The NDPs were dried by lyophilization. The structure of the functionalized NDPs was confirmed by FT-IR spectroscopy.

### 2.2. Method for Determining the Content of Functional Groups on the Surface of a Modified Nanodiamond Powder

#### 2.2.1. Determination of the Amount of Carboxyl Groups

Benzoic acid derivative attached to the surface of the nanodiamond powder (**1**) (5 mg) was weighed into the falcon, and 5 mL of 0.05 M NaOH solution was then added. The resulting suspension was sonicated for 20 min and then stirred for 24 h. After this time, the suspension was centrifuged and 1 mL of solution was taken (triplicate) and diluted to 20 mL of distilled water. After adding 2 drops of methyl orange solution in ethanol, the mixture was titrated with 0.05 M HCl. As the reference point, solely 0.05 M NaOH (5 mL) solution was used, which was titrated with 0.05 M HCl in the same manner as for the samples tested.

#### 2.2.2. Determination of the Amount of Amine Groups

Aminonaphtalene derivative attached to the surface of the nanodiamond powder (**2**) (5 mg) was weighed into the falcon, and 5 mL of 0.05 M HCl solution was then added. The resulting suspension was sonicated for 20 min and then stirred for 24 h. After this time, the suspension was centrifuged and 1 mL of solution was taken (triplicate) and diluted to 20 mL of distilled water. To the solution, 2 drops of phenolphthalein solution in ethanol was added, and the mixture was titrated with 0.05 M NaOH. As the reference point, solely 0.05 M HCl (5 mL) solution was used, which was titrated with 0.05 M NaOH in the same manner as for the samples tested.

## 3. Results and Discussion

### 3.1. Laser Treatment—Introduction of sp^2^ Carbon onto the NDP Surface

The nanodiamond powder, modified using the pulsed laser system, was examined by Raman and FT-IR spectroscopy. Attention was focused on the effect of laser power on graphitization. Figure 2a shows Raman spectra of the NDPs (in a limited range from 950 to 1950 cm^−1^) before and after modification with three increasing values for average laser power. The structure of diamond particles fabricated by the detonation method has been described as an ordered diamond lattice core surrounded by a shell of compressed diamond, which itself is enveloped in a mixture of sp^2^ and sp^3^ bonds [24,25,26]. The Raman spectra of the NDPs before laser treatment confirmed this structure. Bands were observed at around 1200 cm^−1^, attributed to nanocrystaline diamond, at ~1330 cm^−1^, attributed to regular diamond, at ~1420 cm^−1^, attributed to unstructured (disordered) graphite (D band), at ~1550 cm^−1^, attributed to graphite (G band), and at ~1625 cm^−1^, attributed to defects in the structure of the nanocrystalline diamond (grain size <10 nm) [27,28,29,30,31].

Laser processing with an average power of 3.09 W led to the first change in the Raman spectra of the NDPs (Figure 2a). The band at around 1200 cm^−1^ disappeared and others changed their positions. After laser processing at an average power of 8.68 W, a Raman spectrum was obtained with two main features, a broad G band and a broad D band. The absence of a diamond peak at ~1332 cm^−1^ can be explained by the fact that the visible Raman band at around 532 nm is directly sensitive to the presence of sp^2^ bonds in carbon, since this photon energy preferentially excites the π states at these sites [27,28]. This means that graphitization occurs.

Figure 2b shows FT-IR spectra of the NDPs before and after laser treatment. In the spectrum of the unmodified powder, bands can be observed at ~3420 cm^−1^, 1740 cm^−1^, 1630 cm^−1^, and 1140 cm^−1^. The band at ~3420 cm^−1^ corresponds to the stretching (ν_OH_) bond of the OH group, corresponding to water absorption by the powders [30]. The band at ~1740 cm^−1^ can be attributed to the stretching vibration of the carbonyl group (ν_C=O_), while that at ~1140 cm^−1^ corresponds to the stretching vibration of the C–O bond (ν_C-O_) on the surface of the nanodiamond particles. The band at 1630 cm^−1^ corresponds to the stretching (ν_C=C_) double bonds of the carbon atoms on the surface of the NDPs.

Following the modification process, all the functional groups identified in the unmodified powder were visible in the FT-IR spectra (Figure 2b), although their band positions were shifted to lower wavenumbers. The bands for C=C bonds were substantially more intense after laser modification. This result is consistent with the results of Raman spectroscopy, which showed that graphitization on the surface of the particles occurred at higher average laser powers.

To summarize this section of the study, analysis by Raman spectroscopy of the NDPs after the laser treatment confirmed that, regardless of the processing parameters used, the surfaces of the particles were graphitized. The degree of graphitization depended on the processing parameters and increased with the average laser power. However, it was difficult to determine the degree of graphitization precisely because the material after laser treatment also contained unprocessed (primary) NDPs. FT-IR analysis (DRIFTS technique), which was more extensive than the Raman microscope analysis, showed the presence of a band corresponding to stretching C=C bonds in all spectra of the laser modified powders. Compared to the untreated powder, the intensity of the peak for C=C bonds grew steadily, in combination with the closest peak in the spectra, as laser power was increased. The presence of these bands in the FT-IR spectrum suggested the possibility of carrying out further chemical modifications of the nanopowder. For this purpose, an average laser power of 5.90 W was selected.

### 3.2. Chemical Functionalization

Preliminary experiments were conducted with the aim of functionalizing the laser-modified nanodiamonds, using a classical chemical method based on the use of in-situ generated diazonium salts derived from aromatic amines [17]. This method has been successfully used for the modification of thermal annealed nanodiamond surfaces with a variety of aromatic amine derivatives containing azide or propargyl ether residues in the presence isoamyl nitrite [32]. This allows for further functionalization using the click-chemistry method for the conjugation of complex organic moieties [33,34].

Additional modification was attempted using aromatic amines containing additional functional groups: an amine (1,5-diaminonaphtalene) and a carboxylic function (p-aminobenzoic acid) (Figure 3). Use of these substrates allows for the further introduction of functional groups with electrophilic or nucleophilic reagents. Moreover, both groups are present in amino acids/peptides, so they should enable the use of methods typical for peptide synthesis in further downstream functionalization stages.

Based on titration results, it has been found that the NDP loading of the residue of benzoic acid attached on the surface NDP **1** was 0.994 mmol COOH/1 g NDP. In the case of derivative **2**, the loading was 1.426 mmol NH_2_/1 g NDP, respectively.

The reaction products were analysed by means of FT-IR and Raman Spectroscopy (see Figure 4a,b). To enable straightforward interpretation of the FT-IR and Raman spectra of Products **1** and **2**, the spectra of the starting substrates, p-aminobenzoic acid and 1,5-diaminonaphthalene, were also recorded (Figure 4c,d).

In the spectrum of Derivative **1** (functionalization with p aminobenzoic acid), bands are visible characteristic of the carboxylic function. One of these is the wide band of oscillations for the O–H group with maximum absorption in the range of 2500–3500 cm^−1^. The C=O stretching band is also visible. In the case of aromatic acids, the band is in the range of 1680–1700 cm^−1^. An additional characteristic of carboxylic acids is the presence of a band at a wavelength of 900–950 cm^−1^, which corresponds to the deformation vibrations (γ) of the hydroxyl group of the carboxylic function. Modifying NDPs containing double bonds on the surface using p-aminobenzoic acid in the presence of isoamyl nitrite attaches a benzoic acid residue to the surface. Characteristic aromatic bands are also visible in the FT-IR spectra. Aromatic compounds are characterized by the presence of a band from tensile C–H vibrations above 3000 cm^−1^, but these bands are quite often covered by the bands of other groups. Other characteristic bands showing an aromatic structure are the skeleton vibration bands responsible for stretching the C=C bonds in the 1450–1610 cm^−1^ range and C–H deformation bands located at 900 cm^−1^.

The product of modifying olefinated NDPs with 1,5-diaminonaphthalene in the presence of isoamyl nitrite is a derivative containing naphthalenamine residues on the surface. The modified material **2** should show characteristic bands for primary aromatic amines. For amines, the characteristic bands are in the regions 3300–3500 cm^−1^, 1500–1650 cm^−1^, and 1000–1360 cm^−1^. They are related to the vibrations of N–H and aromatic C–H bonds. Primary amines are characterized by the presence of two bands in the 3300–3500 cm^−1^ range, which correspond to asymmetric and symmetrical N–H stretching vibrations. The band in the range of 1500–1650 cm^−1^ is characteristic of N–H deformation vibrations. In addition, the primary amines have a broad absorption band in the range of 660–910 cm^−1^, which corresponds to the N–H fan vibration. The C–N stretching is above 1000 cm^−1^. The other aromatic strands discussed above are also visible.

Comparing the Raman and FT-IR spectra before and after the attachment of modifiers to the functionalized NDPs **2** does not enable identification of typical bands for the modifiers (see Figure 4d). Bands for the NDPs before chemical modification occur in the same places. However, the effect of modification on the shape of the final spectrum can be observed. The sharp peaks of p-aminobenzoic acid (Figure 4d) lead to a spectrum in which it is easy to recognize bands for pure and laser-treated NDPs (see Figure 2b), whereas the wider (at the base) bands for 1,5-diaminonaphtalene (Figure 4d) lead to a smoothed spectrum typical for diamond-like carbon materials.

In the next stage of our research, Derivatives **1** and **2** were used for further functionalization. To Derivative **1** was attached methyl ester of 6-aminohexanoic acid (aminocaproic acid), which possesses hemostatic properties. Derivative **2** was coupled with Ketoprofen, a non-steroidal anti-inflammatory drug. These studies were conducted to examine the possibility of further functionalization of the modified NDPs by attaching compounds with predetermined biological activities. In both cases, 4-toluenosulphonate 4-(4,6-dimethoxy-1,3,5-triazin-2-yl)-4-morpholine (DMT/NMM/TosO^−^) was used as a coupling reagent [35]. In addition to high efficiency, DMT/NMM/TosO^−^ is characterized by the easy removal of side products, thereby ensuring that there were no deposits on the functionalized nanomaterials [36] (see Figure 5).

The structures of the obtained Derivatives **3** and **4** were confirmed based on FT-IR and Raman spectra (Figure 6 and Figure 7).

By comparing the FT-IR spectra for Substrate **1** and Product **3** obtained by coupling with methyl ester 6-aminohexanoic acid, we can evaluate the accuracy of the transformations. It can be observed that bands originating from the hydroxyl group of the carboxyl function of the substrate disappear, while bands originating from the amide formed in the reaction appear. A band for the ester carbonyl group is also visible at 1720 cm^−1^ (Figure 6a, lower panel). The characteristic band at 3400–3500 cm^−1^ in the spectrum of the product corresponds to N–H tensile vibrations. The band at 1630–1680 cm^−1^ corresponds to tensile vibrations in the amide C=O group. A band for Amide I is also visible at the wavelength of 1620–1655 cm^−1^, and the N–H deformation band (δ) at 1510–1550 cm^−1^ corresponds to Amide II. At 1200–1300 cm^−1^, a C–N stretching band (ν) is also visible. The band at 600–800 cm^−1^ corresponds to the deformation vibrations (γ) of the N–H group.

The effect of modification of substrate **1** by methyl ester 6-aminohexanoic acid is not visible in the Raman spectrum (Figure 6b). The spectrum of the final product of chemical modification **3** looks almost the same as that of Substrate **1** and confirms only the presence of the NDPs used as the base for chemical modification.

Comparison of the FT-IR spectra of Substrate **2** with the product of Coupling **4** with Ketoprofen again indicates that the reaction was successful. The characteristic disappearance of the bands derived from the amino group and the appearance of strong characteristic bands at 3400 cm^−1^ derived from the amide formed in the reaction correspond to the results given above. Bands in the range 1690–1710 cm^−1^ characteristic for C=O stretching vibrations in ketones are also visible in the spectra. These results contrast with those from Raman spectroscopy. Coupling of Ketoprofen to Substrate **2** had no influence on the shape of the Raman spectrum of the chemically modified product (Figure 7). This confirms that FT-IR spectroscopy is a more suitable tool for monitoring the structural changes that occur during chemical modification.

In the last stage of our research, we investigated whether the method of functionalizing carbon nanomaterials using aromatic amines and isoamyl nitrite could be used with isokinetic mixtures of two structurally different substrates of nucleophilic and/or electrophilic character, leading to new carbon nanomaterials which can be defined as amphiphilic carbon nanomaterial. To our knowledge, this is the first example of the approach allowing the simultaneous introduction of substituents of opposite nature. Examples of the use of mixtures of aromatic amines and isoamyl nitrite as substrates for the modification of nanomaterials are known in the literature, except that aromatic derivatives containing only uniform electrophilic substituents have been used [32].

Success would allow the simultaneous introduction of structurally differentiated aromatic compounds onto the surface of the carbon nanomaterials. p-aminobenzoic acid and 1,5-diaminonaphthalene were used in the studies. The final product of the reaction was expected to be carbon nanomaterial containing both carboxyl and amine functions, which would allow further selective functionalization of the surfaces of the material (Figure 8).

Based on titration results, it has been found that the loading of residues of benzoic acid and naphthylamine on the surface of NDP **5** was 0.951 mmol COOH/1 g NDP and 1.349 mmol NH_2_/1 g NDP, respectively.

As shown in Figure 9, the FT-IR spectra (Figure 9a) and Raman spectra (Figure 9b) of NDPs modified with an isokinetic mixture of 1,5-diaminonaphthalene and p-aminobenzoic acid contained bands derived from both benzoic acid and naphthylamine. The presented spectra of the modified carbon nanomaterials constitute almost the sum of the bands visible on the spectra for Derivatives **1** and **2**, modified by single components of the isokinetic mixture.

Attempts have also been made to evaluate the effect of NDPs modification on the morphology of the material using SEM microscopic evaluation. The studies used olefinized NDP obtained by laser processing and Derivative **5**, the product of reaction with an isokinetic mixture containing 1,5-diaminonaphthalene and p-aminobenzoic acid. The point of reference was the not modified nanodiamond powder. Figure 10 shows microscopic pictures of the mentioned materials.

As can be seen, the simultaneous introduction of the rest of benzoic acid and naphthylamine onto the surface of the NDP not only changed the morphology of the material, but also made the NDP derivative to be characterized by a lower tendency to agglomerate.

The presence of the same or almost the same amount of carboxyl and amino groups on the surface of the nanomaterial allows its use, under typical conditions used in peptide chemistry, for further modification. In addition, this approach ensures further functionalization under mild reaction conditions without the need for additional deprotection reactions and the use of reagents that could adversely affect sensitive biologically active compounds. Derivative **5** has been used to obtain NDP Material **6** containing a biologically active derivative on the surface as well as a fluorescent label. In studies on surface-modified derivatives (**5**), the folic acid and the 5(6)-carboxyfluorescein derivatives (Figure 11) were attached.

The folic acid residue can be used as a homing molecule with respect to cancer cells, since for many tumors the overexpression of folate receptors is observed. On the other hand, the fluorescent marker allows the tracking of the distribution and interaction of the modified nanomaterial with cells. A triazine condensing reagent (DMT/NMM/BF_4_^−^ and DMT/NMM/TosO^−^), a typical reagent used in the synthesis of peptides, was applied for the synthesis of Derivative **6**. Extremely mild coupling reaction conditions (room temperature, no harsh reagents) have been applied for synthetic purposes, which guarantees the preparation of non-degraded nanomaterials, and it is also crucial from the point of view of sensitive biologically active compounds. The accuracy of synthesis and incorporation of both biologically active derivatives on the surface of the modified Derivative **5** was confirmed based on FT-IR analysis (Figure 12).

By comparing the FT-IR spectra of Substrate **5** and Product **6** obtained via coupling with 5(6)-carboxyfluorescein derivative FL-CONH-(CH_2_)_2_-NH_2_ and folic acid derivative FA-CONH-(CH_2_)_5_-COOH, we can evaluate the accuracy of the transformations (Figure 12a). The characteristic band at 2800–3000 cm^−1^ corresponds to stretching vibrations of the CH group. In addition, the CH group deformation vibration (γ) and twisting vibration (τ) were observed, respectively, at 1370–1390 cm^−1^ and 600–650 cm^−1^. The bands characteristic for the stretching vibrations of C=C were identified at 1590–1630 cm^−1^, whereas the bands characteristic for C=O stretching vibrations occur in the range 1130–1170 cm^−1^. The bands at 1650–1680 cm^−1^ and 780–810 cm^−1^ correspond, respectively, to deformation vibration (δ) and wagging vibrations (ω) of N–H groups. Figure 12b shows Raman spectra of NDPs **6**. The florescent character of used modifier forced a significant reduction in laser power during Raman measurements in order to obtain these spectra. The absence in this spectrum of bands characteristic for NDPs should be considered as another confirmation that the modification has taken place.

## 4. Conclusions

The results of this study show that, by the laser microplasma treatment of NDPs, we can obtain particles with sp^2^ carbon on their surfaces. The intensity of the olefinization process is scalable and depends on the energy of the laser pulse. The C=C bonds on the surfaces of the NDPs enable their use as a substrate in classical chemical functionalization, based on the use of in-situ generated diazonium salts derived from aromatic amines. We were then able to synthesize modified NDPs using 1,5-diaminonaphthalene and p-aminobenzoic acid. The presence of a residue of benzoic acid and naphthylamine on the surface of the laser-modified NDPs was confirmed by FT-IR and Raman spectroscopy. The derivatives were used successfully in further stages of nanomaterial modification using biologically active molecules (Ketoprofen and a 6-aminohexanoic acid derivative). It was also shown to be possible to use isokinetic mixtures of two different aromatic amines containing functional group of opposite character amino and carboxyl group (electrophilic and nucleophilic namely) for functionalization of laser-treated NDPs with sp^2^ carbon on their surfaces. The loading of residues of benzoic acid and naphthylamine on the surface of NDP **5** was 0.951 mmol COOH/1 g NDP and 1.349 mmol NH_2_/1 g NDP, respectively. This approach leads to modified NDPs with amphiphilic properties and allows the use two independent procedures for introducing two different biologically active derivatives to their surface. It has been shown that it was possible to use a modified derivative (**5**) to introduce: (1) a folic acid residue (FA-CONH-(CH_2_)_5_-COOH), a molecule that determines the selectivity of interaction with tumor cells, and (2) a fluorescent marker (FL-CONH-(CH_2_)_2_-NH_2_), which allows one to track modified nanomaterials within in vitro or even in vivo studies, using biocompatible linkers with reduced sensitivity to enzymatic degradation and metabolic processes, leading finally to Derivative **6**. The proposed solution allows simultaneous orthogonal functionalization of the nanomaterial in a one-stage process. A variety of substituents (markers, drugs, homing ligands, etc.) can then be attached to their surfaces.

## Figures and Tables

**Figure 1 nanomaterials-08-00908-f001:**
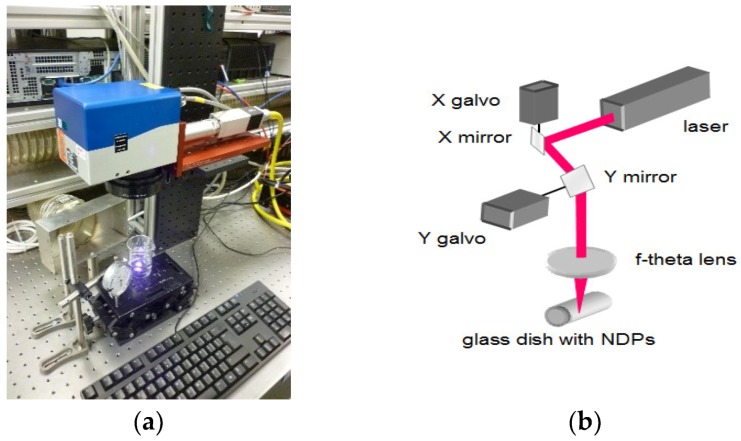
Work station for laser modification of nanodiamond: (**a**) picture of the system (**b**) scheme of the scanner head.

**Figure 2 nanomaterials-08-00908-f002:**
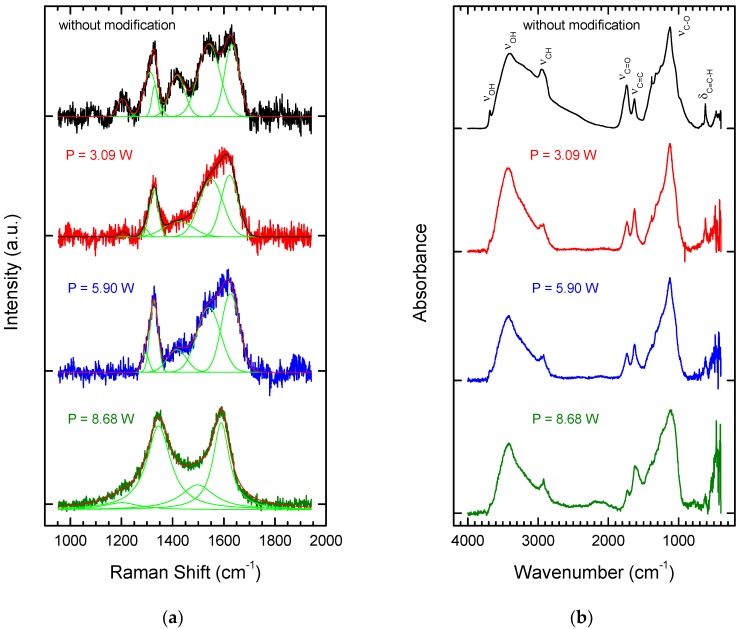
Raman (**a**) and FT-IR spectra (**b**) of nanodiamond particles (NDPs) before and after laser treatment conducted with three average values for laser power.

**Figure 3 nanomaterials-08-00908-f003:**
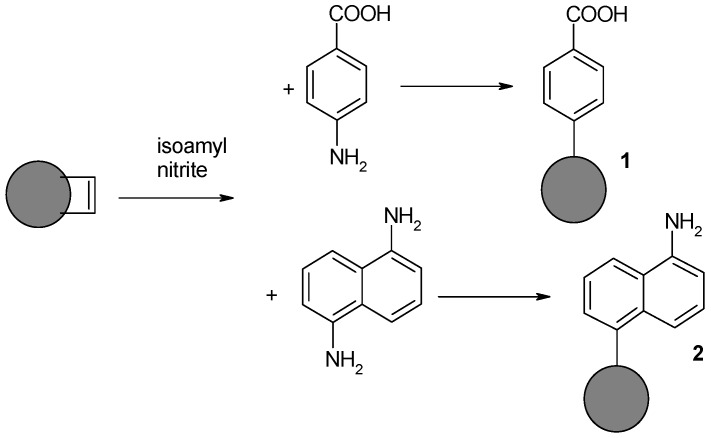
Synthesis of modified NDPs containing residues of benzoic acid and naphthylamine.

**Figure 4 nanomaterials-08-00908-f004:**
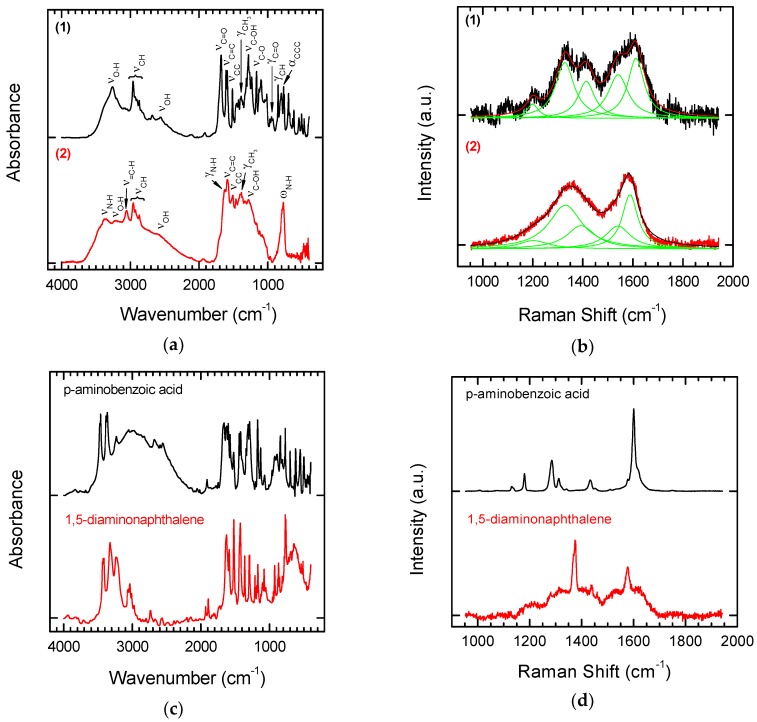
FT-IR (**a**) and Raman spectra (**b**) of laser treated NDPs after chemical modification with p-aminobenzoic acid (upper panel) or 1,5-diaminonaphtalene (lower panel). FT-IR (**c**) and Raman spectra (**d**) of p-aminobenzoic acid (upper panel) and 1,5-diaminonaphtalene (lower panel) used as starting materials for the modification of NDPs.

**Figure 5 nanomaterials-08-00908-f005:**
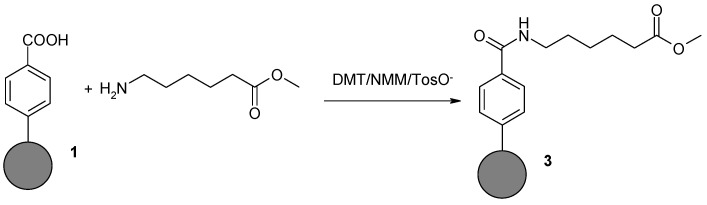
Reaction of modified NDPs **1** and **2** with methyl ester of 6-aminohexanoic acid and Ketoprofen using DMT/NMM/TosO^−^ as a coupling reagent.

**Figure 6 nanomaterials-08-00908-f006:**
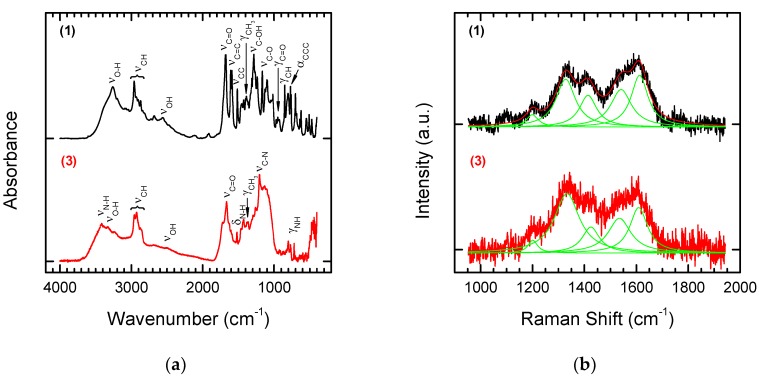
FT-IR (**a**) and Raman spectra (**b**) of NDPs **1** (upper panel) and after coupling with methyl ester of 6-aminohexanoic acid **3** (lower panel).

**Figure 7 nanomaterials-08-00908-f007:**
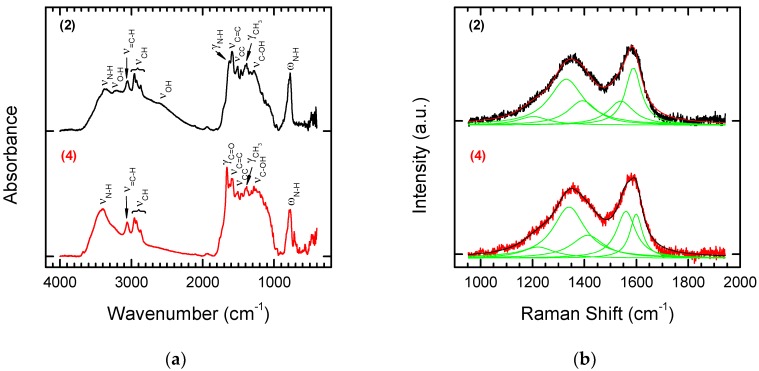
FT-IR (**a**) and Raman spectra (**b**) of NDP **2** (upper panel) modified with Ketoprofen (lower panel).

**Figure 8 nanomaterials-08-00908-f008:**
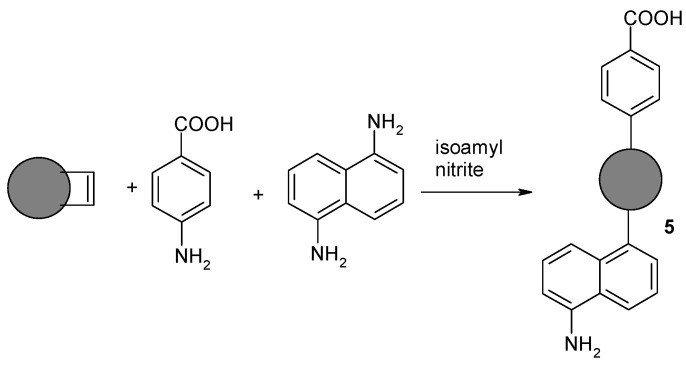
Reaction of olefinated NDPs with an isokinetic mixture of 1,5-diaminonaphthalene and p-aminobenzoic acid.

**Figure 9 nanomaterials-08-00908-f009:**
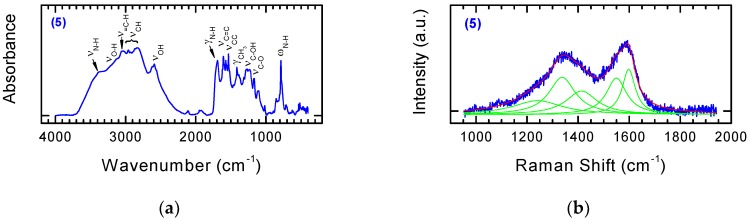
FT-IR (**a**) and Raman spectra (**b**) of NDPs modified with an isokinetic mixture contained 1,5-diaminonaphthalene and p-aminobenzoic acid.

**Figure 10 nanomaterials-08-00908-f010:**
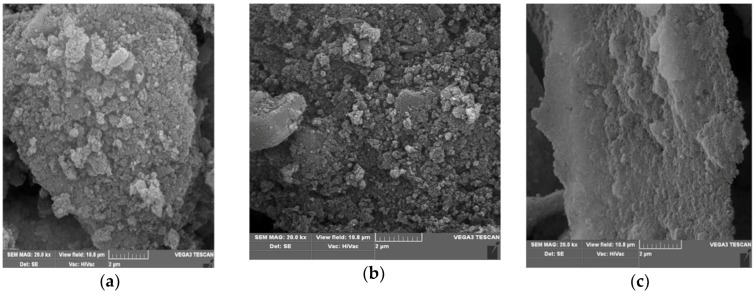
SEM microscopic pictures: (**a**) non-treated NDP (starting material); (**b**) olefinized NDP derivative obtained by laser modification; (**c**) NDP **5** chemically modified with an isokinetic mixture contained 1,5-diaminonaphthalene and p-aminobenzoic acid.

**Figure 11 nanomaterials-08-00908-f011:**
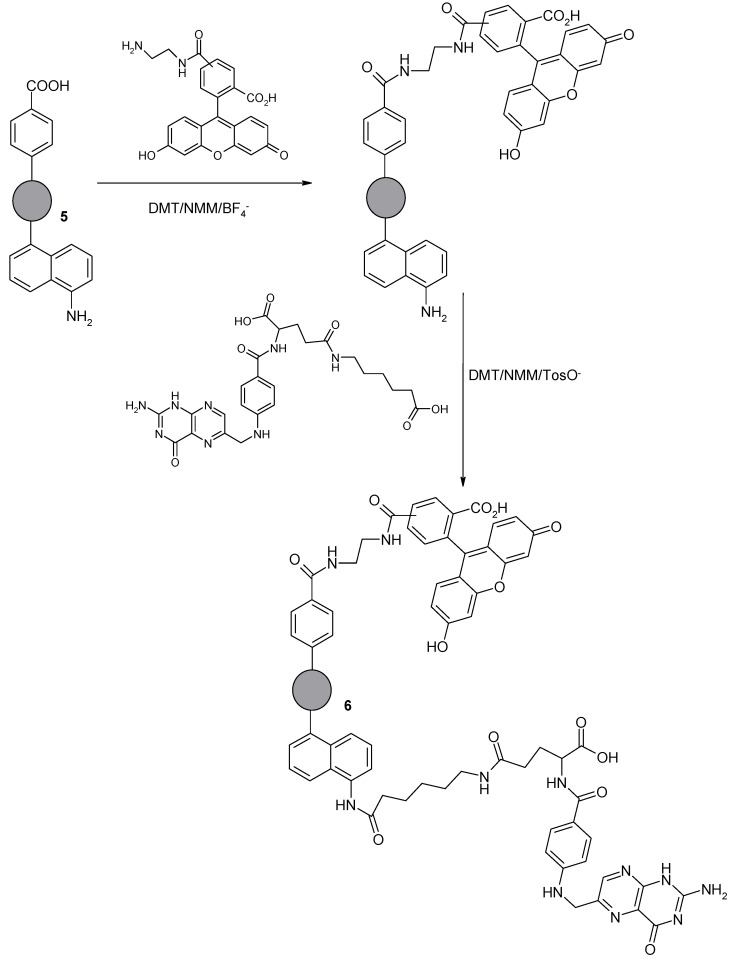
Reaction of the modified NDPs (**5**) with folic acid and 5(6)-carboxyfluorescein derivatives.

**Figure 12 nanomaterials-08-00908-f012:**
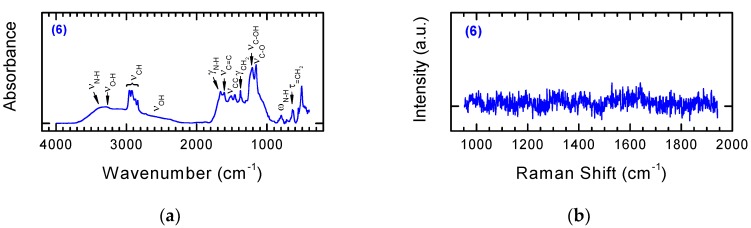
FT-IR (**a**) and Raman spectra (**b**) of NDPs **6** modified with 5(6)-carboxyfluorescein derivative FL-CONH-(CH_2_)_2_-NH_2_ and folic acid derivative FA-CONH-(CH_2_)_5_-COOH.

**Table 1 nanomaterials-08-00908-t001:** Parameters of laser system used during nanodiamond particle (NDP) treatment process.

Parameters	Values
Wavelength	1064 nm
Pulse frequency	25 kHz
Pulse duration	200 ns
Average power	3.09, 5.90 and 8.68 W
Spot size	30 µm
Work area	14 × 6 mm
Hatching distance between parallel lines (lines along the long side of work area)	30 µm
Scanning speed	600 mm/s

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
