# Peer review of "Orthogonal Functionalization of Nanodiamond Particles after Laser Modification and Treatment with Aromatic Amine Derivatives"

_nanomaterials, 2018, doi:10.3390/nano8110908_

Round 1

Reviewer 1 Report

Orthogonal functionalization of nanodiamond particles after laser modification and treatment with aromatic amine derivatives

Justyna Fraczyk , Adam Rosowski , Beata Kolesinska , Anna Koperkiewicz , Anna Sobczyk-Guzenda , Zbigniew Kaminski , Mariusz Dudek

Herewith I am submitting my reviewer comments for the above-mentioned manuscript, which is under consideration to be published in Nanomaterials.

The article is about a new way to functionalise nanodiamonds particles to achieve orthogonality. The synthesis consists of graphitization by a laser followed by functionalization by aromatic amines (single compounds or mixtures are possible) using a reaction that goes via diazonium salts. The steps after the laser modification have already been shown for nanodiamonds from a different synthesis route by thermal annealing. The present approach might offer a bit more flexibility by tuning the laser power. Overall, this is a rather straight forward synthesis approach that leads to an interesting nanomaterial. The effectiveness of the process of laser treatment and chemical modification was monitored using Raman and FT-IR spectroscopy. The article is purely chemical and the authors do not show any biological data. Due to the simplicity and elegance of the approach this is nevertheless interesting.

Page 2 line 87: “This approach would allow the simultaneous addition of aromatic derivatives with different functional groups in the same and/or similar concentration on the surfaces of the NDPs,” Is there a reason why you narrow the scope to same and/or similar concentrations? Wouldn’t it also work with very different concentrations? (differences in reactivity will certainly influence the ratio of molecules that bind but that doesn’t necessarily mean that it wouldn’t work) Or is there another reason for this statement?

Page 3 line 101: “The diameter of the laser spot on the substrate was about 30 um.” What does that tell about the throughput of the method? Also the laser only influences the surface of the powder doesn’t it? So I assume the powder can be a thin film at max. Or does it even need to be single dispersed particles? How much material can you make per hour/min…?

Page 6 line 224: The verb is missing in the sentence starting with “The nanodiamond powders modified using”

Page 12 line 388: “Attempts have also been made to evaluate the effect of NDPs modification on the morphology of the material using SEM microscopic evaluation.” The SEM shows massive aggregates (for all materials). Is that an artifact from drying or was there never an effort to deaggregate the particles? Does the method have an effect on the aggregation behavior? The aggregation state would be the most interesting properties one could see in SEM. So it would be better to have SEM where you see separated particles (unless the method produces aggregates, but that would be a serious drawback that should be more explicitly mentioned)

Page 13 line 397: “but also made the NDP derivative to be characterized by a lower tendency to agglomerate” Maybe I am reading the SEM wrongly but to me they all look massively aggregated.

Page 13 line 401: “this approach ensures full biocompatibility without the need for additional deprotection reactions and the use of reagents that could adversely affect sensitive biologically active compounds.” To my opinion this claim is not justified. Unfortunately, biology is not that predictable. Also biocompatibility for what? To justify the claim this actually needs to be shown. Or at least toned down.

Page 15 line 447: “allows the use of biocompatible procedures for introducing two different biologically active derivatives to their surface” What is meant by biocompatible here? As I read it I think what is meant is that the conditions do not degrade the biomolecule that you want to attach but I wouldn’t call that biocompatibility. Please specify.

Author Response

Concern: Manuscript ID: nanomaterials-382899 "Orthogonal functionalization of nanodiamond particles after laser modification and treatment with aromatic amine derivatives" submitted to Nanomaterials

Dear Sir,

I appreciate very much the comments given by You and Reviewer. In response to suggestions the following revision was introduced (marked in text by yellow background):

Reviewer 1:

1) Page 2 line 87: “This approach would allow the simultaneous addition of aromatic derivatives with different functional groups in the same and/or similar concentration on the surfaces of the NDPs,” Is there a reason why you narrow the scope to same and/or similar concentrations? Wouldn’t it also work with very different concentrations? (differences in reactivity will certainly influence the ratio of molecules that bind but that doesn’t necessarily mean that it wouldn’t work) Or is there another reason for this statement?

In the studies we would like to check whether the use of two different aromatic amines (with a carboxylic and amino substituent) in the same concentrations at the same time will allow to obtain modified materials containing the same (or at least similar) amounts of introduced substituents. There was a risk that the diverse nature of the substituents (electron-withdrawing and electron-donor group) in aromatic rings could affect the efficiency of the modification, which could depend on the nature of the substituents. However, we have found that this effect can be neglected, so it is possible to use differentiated isokinetic mixtures in reactions. It seems that it is also possible to use different concentrations of substrates, which should lead to the modulation of the amount of functional groups introduced on the surface of the nanomaterial.

2) Page 3 line 101: “The diameter of the laser spot on the substrate was about 30 um.” What does that tell about the throughput of the method? Also the laser only influences the surface of the powder doesn’t it? So I assume the powder can be a thin film at max. Or does it even need to be single dispersed particles? How much material can you make per hour/min…?

Proper selection of average power during laser treatment process allows on modification only surface of NDPs. In this paper we focuses on indicate new, more flexible way of treatment of NDPs surface, and indicate of advantages of this process in the case of chemical modification process. Currently, we have not considered how much material can be treated per hour/min. However a method is flexible and by proper modification of laser system large amount of powders can be treated.

3) Page 6 line 224: The verb is missing in the sentence starting with “The nanodiamond powders modified using”

The sentence has been corrected.

4) Page 12 line 388: “Attempts have also been made to evaluate the effect of NDPs modification on the morphology of the material using SEM microscopic evaluation.” The SEM shows massive aggregates (for all materials). Is that an artifact from drying or was there never an effort to deaggregate the particles? Does the method have an effect on the aggregation behavior? The aggregation state would be the most interesting properties one could see in SEM. So it would be better to have SEM where you see separated particles (unless the method produces aggregates, but that would be a serious drawback that should be more explicitly mentioned)

We did not undertake any attempts to de-aggregate of nanomaterial in the study. We only would like to check if the next steps in the functionalization of the NDPs surface would increase the susceptibility to aggregation of molecules. SEM images show that the final modification products using an isokinetic mixture of aromatic amines are characterized by a lower tendency to aggregate, which is therefore beneficial from the point of view of using this method to introduce biologically active compounds on the surface of nanomaterials.

5) Page 13 line 397: “but also made the NDP derivative to be characterized by a lower tendency to agglomerate” Maybe I am reading the SEM wrongly but to me they all look massively aggregated.

Indeed, SEM images show NDPs agglomerates. However, compared to the initial NDPs (commercially available or after the olefinization step) chemically modified nanomaterials were characterized by the presence of a smaller amount of "compact, agglomerated" fraction and were more "homogeneous".

6) Page 13 line 401: “this approach ensures full biocompatibility without the need for additional deprotection reactions and the use of reagents that could adversely affect sensitive biologically active compounds.” To my opinion this claim is not justified. Unfortunately, biology is not that predictable. Also biocompatibility for what? To justify the claim this actually needs to be shown. Or at least toned down.

The text has been corrected and the term 'biocomatibility' has been deleted. It has been used in the sense of the possibility of using modified NDPs to introduce biologically active compounds that are sensitive to the drastic conditions of chemical reactions. The incorporation on the surface of the modified NDPs of compounds (including peptides) that affect interaction with key compounds present in the cell should fulfill the biocompatibility requirement.

7) Page 15 line 447: “allows the use of biocompatible procedures for introducing two different biologically active derivatives to their surface” What is meant by biocompatible here? As I read it I think what is meant is that the conditions do not degrade the biomolecule that you want to attach but I wouldn’t call that biocompatibility. Please specify.

Yes, indeed, the term "biocompatible" was used unnecessarily. In the corrected manuscript, it has been removed.

Sincerely Yours,

Mariusz Dudek

Reviewer 2 Report

The manuscript describes investigation on functionalization of nanodiamonds prodcued by detonation method. By having moderate novelty it may be still interesting for readers because of extendent description of experimental results.

There are few points requiring modifications:

page 1, line 39 - remove '- the detonation method [1]-'

Number of references indicated in Introduction should be significantly reduced (up to 1 review. May be [6] will be enought. Other refs are not used in the text and thus these citations are not necessary.

page 2, line 48 'topologies' should be replaced by 'atomic arrangement' or similar

page 2, line 58 - clarification or appropriate reference is necessary for statement '...substantially higher reactivity of sp2 carbon compared to sp3 carbon...'

page 2, line 72 - misprint in ultrasoni[fi]cation

page 3, line 94 - please explain what does it means 'purity above 98%'; is it purity in respect of non-carbon contaminations (probably YES) or non-diamond?

page 4, line 131 - how thick the gold layer was? If seems that this methodology is not suitable for this particular application because it does not provide reasonable possibility for observation of morphology of nano-powdered samples.SEM images presented in Fig. 10 indicate no significante difference for the samples and thus these techniques is not informative. It may be more reasonbale to use usual optical microscopy without gold layer deposition.

page 6, lines 230-234 - attribution of observed Raman lines to different carbon species is not perfect. For example, why width of 'diamond' line at 1330 1/cm is so large? D-band is usually assigned to 1350 1/cm line of disordered graphite, while, for 'normal' graphite, line G is at 1580 1/cm. This description should be improved and suitable references are provided.

page 8, fig. 4 - why signal noise for Raman spectra shown in the panels (b) and (d) so much different? Is it means difference in material amount/concentration? How different are investigated samples?

Author Response

Concern: Manuscript ID: nanomaterials-382899 "Orthogonal functionalization of nanodiamond particles after laser modification and treatment with aromatic amine derivatives" submitted to Nanomaterials

Dear Sir,

I appreciate very much the comments given by You and Reviewer. In response to suggestions the following revision was introduced (marked in text by yellow background):

Reviewer 2:

1) page 1, line 39 - remove '- the detonation method [1]-'

The sentence has been corrected.

2) Number of references indicated in Introduction should be significantly reduced (up to 1 review. May be [6] will be enought. Other refs are not used in the text and thus these citations are not necessary.

The manuscript has been corrected. The number of the references has been significantly reduced.

3) page 2, line 48 'topologies' should be replaced by 'atomic arrangement' or similar

The sentence has been corrected.

4) page 2, line 58 - clarification or appropriate reference is necessary for statement '...substantially higher reactivity of sp2 carbon compared to sp3 carbon...'

The sentence has been corrected.

5) page 2, line 72 - misprint in ultrasoni[fi]cation

It has been improved.

6) page 3, line 94 - please explain what does it means 'purity above 98%'; is it purity in respect of non-carbon contaminations (probably YES) or non-diamond?

The sentence has been supplemented about explanation: low contents of metallic contamination.

7) page 4, line 131 - how thick the gold layer was? If seems that this methodology is not suitable for this particular application because it does not provide reasonable possibility for observation of morphology of nano-powdered samples. SEM images presented in Fig. 10 indicate no significante difference for the samples and thus these techniques is not informative. It may be more reasonbale to use usual optical microscopy without gold layer deposition.

Yes, indeed, the SEM images presented in Fig. 10 indicate no significante difference for the samples. We only would like to check if the next steps in the functionalization of the NDPs surface would increase the susceptibility to aggregation of molecules. SEM images show that the final modification products using an isokinetic mixture of aromatic amines are characterized by a lower tendency to aggregate, which is therefore beneficial from the point of view of using this method to introduce biologically active compounds on the surface of nanomaterials.

8) page 6, lines 230-234 - attribution of observed Raman lines to different carbon species is not perfect. For example, why width of 'diamond' line at 1330 1/cm is so large? D-band is usually assigned to 1350 1/cm line of disordered graphite, while, for 'normal' graphite, line G is at 1580 1/cm. This description should be improved and suitable references are provided.

Description of bands in Raman spectra of NDPs based on reference (added to text of manuscripts):

27. Dillon, R.O.; Wollam, J.A.; Katkanant, V. Use of Raman scattering to investigate disorder and crystallite formation in as-deposited and annealed carbon films. Physical Review B 1984, 29, 3482-3489, DOI: https://doi.org/10.1103/PhysRevB.29.3482.

28. Tsai, H.; Bogy, D.B. Characterization of diamondlike carbon films and their application as overcoats on thinfilm media for magnetic recording. J. Vac. Sci. Technol. A 1987, 5, 3287-3313, DOI: https://doi.org/10.1116/1.574188.

29. Schwan, J.; Ulrich, S.; Batori, V.; Ehrhardt, H.; Silva, S.R.P. Raman spectroscopy on amorphous carbon films, J. Appl. Phys. 1996, 80, 440-447, DOI: https://doi.org/10.1063/1.362745.

30. Mochalin, V.; Osswald, S.; Gogotsi, Y. Contribution of Functional Groups to the Raman Spectrum of Nanodiamond Powders. Chem. Mater. 2009, 21, 273-279, DOI: 10.1021/cm802057q.

31. Korepanov, V.I.; Hamaguchi, H.; Osawa, E.; Ermolenkov, V.; Lednev, I.K.; Etzold, B.J.M.; Levinson, O.; Zousman, B.; Epperla, Ch.P.; Chang, H.-Ch. Carbon structure in nanodiamonds elucidated from Raman spectroscopy, Carbon 2017, 121, 322-329, DOI: http://dx.doi.org/10.1016/j.carbon.2017.06.012.

Particle size of NDPs was about 4 nm. So diamond line is not intense and narrow, but is low and broad. D-band is usually assigned to 1350 cm-1, but also in the case of NDPs is observed shift of this band to higher value of frequency (G-band shift to lower value) and this situation is observed in our case.

9) page 8, fig. 4 - why signal noise for Raman spectra shown in the panels (b) and (d) so much different? Is it means difference in material amount/concentration? How different are investigated samples?

Figure caption explain difference between panels (b) and (d). Figure 4d shows Raman spectra of substrates attached to NDPs during chemical modification, whereas figure 4b product of this modification.

Sincerely Yours,

Mariusz Dudek

Reviewer 3 Report

The paper entitled "Orthogonal functionalization of nanodiamond particles after laser modification and treatment with aromatic amine derivatives” by Fraczyk et al. deals with the laser modification of diamond nanoparticles with a 1 um laser and the functionalization of the treated nanoparticles using aromatic amines. This leads to amphiphilic carbon nanoparticles which can serve to incorporate useful biological molecules. Raman, and FTIR confirmed the graphitization of the surface of the nanoparticles.

The results are interesting, but after reading the paper, I have some comments about it:

COMMENTS:

1) This paper is well-written and only requires minor modifications. Has been measured the average particle diameter after the laser treatment? Probably, the particles are smaller after the laser treatment.

2) (Page 2) Please, replace “laser mashing” with “laser ablation”. This kind of process is conventionally referred as ablation.

3) (Page 3) Where was the focus spot positioned?

4) (Page 3) Please, give the overlapping between scan lines.

5) (Page 7) Was the extension of the graphitization measured for individual particles? Is graphitized the whole particle or only its surface?

Author Response

Concern: Manuscript ID: nanomaterials-382899 "Orthogonal functionalization of nanodiamond particles after laser modification and treatment with aromatic amine derivatives" submitted to Nanomaterials

Dear Sir,

I appreciate very much the comments given by You and Reviewer. In response to suggestions the following revision was introduced (marked in text by yellow background):

Reviewer 3:

1) Has been measured the average particle diameter after the laser treatment? Probably, the particles are smaller after the laser treatment.

Particle size of nanodiamond powder delivered by Sigma Aldrich Company was measured by Transmission Electron Microscope and confirmed by X-Ray Diffraction measurements. After laser ablation process we focus on analysis of NDPs using Raman and FT-IR spectroscope. Inspected decreased size of diamond root of particle was not monitored by XRD measurements. We select Raman spectroscopy as the best tool for analysis of surface change (graphitization) in treated powders, and FT-IR spectroscopy as tool for monitoring of chemical modification.

2) (Page 2) Please, replace “laser mashing” with “laser ablation”. This kind of process is conventionally referred as ablation.

It has been corrected.

3) (Page 3) Where was the focus spot positioned?

Focus spot during laser treatment process was positioned on surface of powder spilled in glass ampoule placed on table of laser system – see figure 1b. This information was added to manuscript.

4) (Page 3) Please, give the overlapping between scan lines.

According with information shows in Table 1 (scan speed: 600 mm/s and estimated impulse speed: 30 mm ´ 25 kHz = 750 mm/s) overlapping was equal 20%.

5) (Page 7) Was the extension of the graphitization measured for individual particles? Is graphitized the whole particle or only its surface?

Information about graphitization of NDPs during laser treatment process based on Raman and FT-IR spectroscopy measurements. According to results shows in Figure 2a (Raman spectra) significant graphitization process – not only on surface – occur in the case of laser treatment process with 8.68 W of average power. Because, the aim of this part of work is preparing NDPs for chemical modification, detail analysis of graphitization process (decrease of size of diamond root of particle) was postponed for future investigation by using TEM or XRD technique.

Sincerely Yours,

Mariusz Dudek

Round 2

Reviewer 2 Report

The revisions made in the manuscript are satisfactory.